# Gene Pool of Winter Wheat from the World Collection of N.I. Vavilov Institute of Plant Industry (VIR) for Biotic Stress Resistance

**DOI:** 10.3390/pathogens10050514

**Published:** 2021-04-23

**Authors:** Sulukhan Temirbekova, Ibrahim Jafarov, Ivan Kulikov, Yuliya Afanaseva, Elena Kalashnikova

**Affiliations:** 1All Russian Research Institute of Phytopathology, Bolshye Vyazyomy, Odintsovo District, 143050 Moscow, Russia; 2Department of Plant Protection, Agrarian University, Ganja AZ2000, Azerbaijan; office@adau.edu.az; 3Federal Horticultural Center for Breeding, Agrotechnology and Nursery, 115598 Moscow, Russia; vstisp@vstisp.org (I.K.); yuliya_afanaseva_90@bk.ru (Y.A.); 4Moscow Timiryazev Agricultural Academy, Agrarian University, 127550 Moscow, Russia; kalash0407@mail.ru

**Keywords:** winter wheat, VIR gene pool, biotic stresses, *Fusarium* blight, *Septoria* blotch, EMDS, root rot, barley yellow dwarf virus

## Abstract

This paper presents the results of the 50 year-long research into the winter wheat gene pool from the VIR world collection in the Moscow region to assess biotic stress resistance following N.I. Vavilov’s concept of the ‘ideal variety’, proposed in 1935. The Federal Scientific Selection and Technology Center for Horticulture and Nursery was responsible for the field studies of winter wheat, and the All-Russian Research Institute of Phytopathology and Russian State Agrarian University—Moscow Timiryazev Agricultural Academy—for phytopathological studies. The wheat collection was studied in compliance with the VIR Methodological Guidelines using the International COMECON list of descriptors for the genus *Triticum* L. Resistance against the enzyme–mycotic depletion of seeds (EMDS) was tested using original techniques. It was found that annual brown rust and powdery mildew attacks in the collection’s winter wheat samples caused no significant economic damage. One case of *Septoria* head and leaf blotch, two cases of *Fusarium* head blight, one case of root rot, one case of barley yellow dwarf virus, 20 cases of EMDS, and three cases of 3rd-degree EMDS, i.e., seed germination in an ear, were recorded. The parent material resistant to the biotic stresses of the region was selected for breeding. Domestic breeders have created outstanding wheat varieties close to the ‘ideal’ as noted by N.I. Vavilov.

## 1. Introduction

N.I. Vavilov recognized the special role of resistance to diseases in producing an ideal cultivar as early as 1935 [1]. He believed that cultivating pest- and disease-resistant plant varieties was among the top priorities of plant breeding. As a pioneer researcher of the theory of plant immunity to infectious diseases, he later became a single author of the genotypic immunity theory, the theoretical backbone of plant introduction. N.I. Vavilov proved to plant breeders, geneticists, and botanists that immunity was linked to plants’ genetic natures, and the responses of host plants to parasite invasions were determined by the host’s genetic taxonomy compared to other close species. The advancements in genotypic immunity theory brought to light some principles of the co-evolution of hosts and parasites in their places of origin that could benefit the plant breeding industry. Today, the most difficult and pressing plant breeding issue is to combine resistances to multiple diseases in one variety. When it comes to breeding for immunity, possible changes in the racial composition of pathogens depending heavily on weather changes and the introduction of new virulent races have to be taken into account [2,3]. Thus, group immunity, i.e., resistance to multiple pathogens of many physiological races, is increasingly important. 

Currently, molecular methods are used to study the gene pool of plant resources from the VIR world collection; for example, Riaz et al. [4] used selected lines from the VIR collection to create a diversity panel of genome-wide association studies (GWAS) to identify new alleles of leaf rust resistance. Such methods as nested-association mapping (NAM) и and multi-parent advanced generation intercross (MAGIC) can also improve the detection of rare alleles [5]. The use of local varieties can significantly increase the diversity and change the frequency of alleles, which will allow us to identify new sources of resistance to pathogens.

In the former Moscow Branch of the VIR named after N.I. Vavilov, now the Department for the Gene Pool and Biological resources of Plants of the Federal Scientific Selection and Technology Center for Horticulture and Nursery, for more than 50 years, the gene pool of the world collection of winter wheat has been studied, which comprises about three thousand samples. 

The goal of the presented study was to select the donor material resistant to multiple diseases to produce winter wheat varieties with group immunity.

## 2. Results and Discussion

Listed below are the sources of soft winter wheat by resistance to the most common diseases identified during epiphytotic years.

### 2.1. Pink Snow Mold, Pathogen Microdochium nivale ((Fr) Samuels and I.C. Hallett) (syn. Fusarium nivale (Fr.) Ces.)

Seven major outbreaks of pink snow mold in winter wheat were reported during the research period in 1985–1986, 1988–1989, 1989–1990, 1997–1998, 2000–2001, 2002–2003, and 2004–2005. *Microdochium nivale* ((Fr) Samuels and I.C. Hallett) was identified as the pathogen by phytopathological examination.

The symptoms of the disease were as follows: plants affected early and severely had distorted, glued together, and dead leaves with whitish-pink taint. The loss of plants resulted in the decimation of crops. The surviving plants had watery spots with whitish-pink weblike taint from mycelium and conidial sporulation. The spots first emerged on lower leaves and gradually turned brown (see Figure 1). The dieback of glued together leaves was followed by the destruction of tillering nodes. In some years, the ascosporulation of fungi in the form of round reddish fruiting bodies (perithecia) could be observed in dead plants by microscopic examination.

In terms of resistance to pink snow mold, the number of grains per ear, grain coarseness, lodging resistance, and crop yields, the following cultivars stood out in 1989–1998: Linna (k-45889), Hja 24499 (k-62273) from Finland; *WW 23262* (k-51808), Holger (k-62310), Sheriff Dickopf (k-40526), Svalofs Sonnet II (k-45132), Sv 61246 (k-47099), WW 24089 (k-51803), Sv 65646 (k-55305), Hildur (k-54130), Kosack (k-58137) from Sweden; PP 114-74 (k-57618), Liwilla (k-57580) from Poland; Hornet (k-60100), Legend (k-61498), Maris Ploughman (k-57944) from the United Kingdom; PE 6490 (k-52656, Denmark); Zdar (k-57255, Czech Republic); Remus (k-56904), Caristerm (k-57610), Tukan (k-57579) from Germany; Raduga (k-50948), Nemchinovskaya 846 (k-56861), Lutestsens 497/83 (k-57657) from the Moscow Region, Russia; Brigantina (k-55181, Ukraine).

In 2003 and 2005, the following cultivars stood out in terms of resistance to pink snow mold and crop yields: reference variety Moskovskaya 39, (crop yield of 178 g/m^2^); Kazanskaya 560 (k-63565, Tatarstan)—327 g/m^2^; k-15339 (Belarus) —295 g/m^2^; Milturum 5811 (k-40710, Leningrad Region)—192 g/m^2^; k-11231 (Voronezh Region)—190 g/m^2^; Ta b 2598, k-44326 (187 g/m^2^); Finnish samples: *B 503* (k-44839)—177 g/m^2^, Antti (k-42673)—150 g/m^2^; Karelskaya bezostaya (k-40579, Karelia)—160 g/m^2^; breeding lines from Krasnodar Krai: KOS 2168-92 (k-63584)—140 g/m^2^, KOS 2113-92 (k-63595)—130 g/m^2^, *KOS 2300-92* (k-63588)—140 g/m^2^; Severodonetskaya yubileinaya (k-63567, Rostov Region)—160 g/m^2^; samples from the United States: Hoff (k-63543)—130 g/m^2^, Hall (k-63556)—160 g/m^2^. The samples listed above had the average numbers of grains per ear of 42–59 with the 1000-grain weight of 45 g.

### 2.2. Powdery Mildew, Pathogen Blumeriagraminis (DC.) Speer (syn. Erysiphegraminis DC.f.sp. tritici Em. Marshal)

The epiphytotic years of powdery mildew were 1981 and 1987. It is worth mentioning that despite the necessity to evaluate powdery mildew resistance in winter wheat from the world collection, no significant damage from the disease and no associated crop losses were reported in the Moscow Region throughout the research period. All samples, both foreign and domestic, showed high tolerance to the disease.

**In 1981**, 162 wheat samples out of 547 caught mild powdery mildew infection (score of 7), 185—moderate and severe infection (scores of five and three), and the reference cultivar Mironovskaya 808 was affected very severely by powdery mildew (score of one). Mild powdery mildew infection was also observed in cultivars and breeding lines Hohenthurmer 6921/68 (k-50620), Hohenthurmer 5171/67 (k-50631), Hohenthurmer 2078/70 (k-50628), Hohenthurmer 20901/70 (k-50629), Hohenthurmer 13653/67 (k-50632), Hohenthurmer 27691/71 (k-50675), Hohenthurmer 29521/69 (k-50676), Hadmerslebener 22228/70 (k-50612) from Germany; Oasis (k-51829), Tecumseh (k-51792) from the United States; Wattines (k-50740, France). In terms of crop yields and 1000-grain weight, the resistant samples (400–450 g/m^2^ and 45–47 g) outperformed the reference cultivar (400 g/m^2^ and 37 g).

**In 1987**, the following varieties out of 500 collection samples showed powdery mildew resistance (score of 7): Kronjuwel (k-57615), Urban (k-59547) from Germany; UN7050 (k-57255, Czech Republic); Sv. 75268 (k-56156), Helge (k-56872), Walde (k-51794), Sv. Vg 74393 (k-56065) from Sweden; FM-187 (i-441409, Poland); Maris Durrin (k-55232), Maris Marksman (k-55233) from the United Kingdom; Kolybelovskaya 33 (k-56422, Samara Region); Zarya (k-49916, Moscow Region). Crop yields of the resistant samples matched those of the reference cultivar, i.e., 550–600 g/m^2^, but their 1000-grain weights were lower at 30–40 g (compared to 41 g for the reference cultivar).

### 2.3. Brown Rust, Pathogen Puccinia triticina Eriks

Mandatory evaluation of brown rust resistance in winter wheat from the VIR gene pool was performed annually. Severe outbreaks in 1976 and 1977 had no significant effect on the wheat samples. Resistant samples underperformed in crop yields and 1000-grain weights by 10–25%, compared to the reference cultivar. However, their high resistance and tolerance to the disease may be considered of higher breeding value. In terms of brown rust resistance, the following varieties stood out: Dippes Triumph (k-45029), Sylvia (k-46607), Ibis (k-45335) from Germany); Sturdy (k-48223), Wester (k-48229), Wesel (k-48231), Kansas 594-2 (k-51242), Nelson (k-63522), Charmany (k-63526), TAM 108 (k-61609), TAM 200 (k-61610), Ks 90 WGRG 10 (k-62377) from the United States) IIves (k-63016, Finland); Justus (k-63280), Leopold (k-63275), Expert (k-63273) from Austria; Alrakis (k-63369, Switzerland); Volzhskaya 6 (k-63120, Ulyanovsk Region); Splav (k-63117), Tau (k-63002) from the Vladimir Region; Moskovskaya 39 (Moscow Region).

The sample Neuzucht 14/14 (k-40109, Germany), used by P.P. Lukyanenko as breeding material in the Moscow Region in the 1990s, became increasingly susceptible to brown rust, possibly due to the emergence of new physiological races of the pathogen.

### 2.4. Root Rots, Fusarium Blight, Pathogen Fusarium sp., Including F.culmorum F.avenaceum F.oxysporum Schlecht. ex Fr., and helminthosporiosis or Common Root Rot, Pathogen Bipolaris sorokiniana Shoem. (syn. Drechslera sorokiniana Subram.et Jain, Helmintosporium sativum Pam.)

A severe outbreak of root rots of mixed etiology was reported in 1990. The disease manifested itself before the milky ripeness stage in the form of white ear rot, a common symptom of root rots. Out of 500 studied samples, 40 produced small shrunken grains viable after reseeding. Since root rots of mixed etiology are harmful and persistent mycoses, phytosanitary monitoring is a necessity [6]. 

Throughout the research period, the collection samples were tolerant to the root rot pathogen, and the disease had no significant effect.

### 2.5. Resistance to Enzyme-Mycotic Depletion of Seeds

Resistance to increased and excessive humidity during a grain filling period causing the development of grain exhaustion or enzyme–mycotic depletion of seeds (EMDS) is considered another limiting factor in the Moscow Region. N.G. Kholodny [7], M.S. Dunin, S.K. Temirbekova [8], and B.I. Sandukhadze [9] noted that the shortage of above-zero temperatures (°C) and increased atmospheric humidity often observed during grain maturation led to a decrease in crop yields (by 30–50% and above) combined with grain quality degradation. 

Throughout the research period, EMDS was reported 21 times in 1976–1980, 1981, 1985, 1986, 1987, 1990, 1991, 1994, 1995, 1999, 2003, 2004, 2006, 2009, 2013, 2019, and 2020 as an associated disease.

The abiotic and biotic causes of the disease and key pathogenetic features (emergence and development) of EMDS may be characterized as associated pathological processes mostly occurring in two stages. In some cases, an additional markedly different third stage, i.e., germination in standing crops, windrows, and wet sheaves [8,10]. 

Throughout the whole research period, abundant precipitation was reported in the blooming or grain filling phase, which favored the development of the first (noninfectious) enzyme stage, i.e., biological injury of standing crops, and mycotic stage (ear disease). The third stage, i.e., germination in standing crops, was reported three times in 50 years.

It is commonly known that favorable conditions for manifestations of abiotic stress activity emerge much earlier than those favoring phytopathogenic activity. Noninfectious enzyme processes under humid conditions produce a perfect nutritious substrate in plants as early as the blooming phase. We define biological injuries in standing crops as macro- and micro-injuries caused by osmotic and hydrostatic pressure at the enzyme stage [11,12] and consider them a gateway for phytopathogen invasion. Earlier, authors only recognized mechanical injuries of seeds during harvesting and drying. We, however, have been the first to discover biological injuries in standing crops and distinguish hidden and open injuries at blooming and full ripeness phases (see Figure 2).

The data on dry matter loss at various maturation phases and 1000-grain weights confirm that each consecutive wetting of ears and grains of wheat and other cereals, such as rye, triticale, barley, oat, and corn, generating a new portion of protein, carbohydrate, and lipid hydrolysis products, thereby forms a perfect environment for phytopathogenic fungi, which boosts their metabolism and leads to the further destruction of seeds. Under these conditions, parasitic forms are selected from populations of saprophytic and semi-parasitic fungi, such as *Alternaria* and *Fusarium*, populating the nutritious environments of varying biochemical quality in the aforementioned cereals. As a result, the viability of the stored seeds is suppressed and the germinating rate of seeds planted in lean years is reduced. The harmfulness of seed infection increases from the time of infection to seedling formation. After that, the soil mycobiota plays a major part in the pathogenic processes.

Massive outbreaks of alternariosis with pathogen *Alternaria alternata* (Fries) Keisler (syn. *A. tenuis* Nees.) were observed in wheat, rye, and winter triticale ears in the Moscow Region 17 times in a 21 year period (see Figure 3). 

In some years, the ecological niche of alternariosis in ears and grains was occupied by *Fusarium* blight, and in 1987 by septoria blight of leaves and ears. Our observations showed that fungal spores in cereals emerged at the earing and blooming phases [8], and their numbers increased sharply at the grain filling phase. The genus *Alternaria* was almost universal and well-adapted fungi populating reproductive organs (anthers) and caryopses of wheat, oat, barley, rye, and triticale in the Central Region of Russia. *Alternaria* infected ears and seeds under humid conditions from blooming to harvesting phases and during storage. To grow and reproduce, the fungi used the perfect nutritious substrate available due to the hydrolysis of metabolic products of the plants emerging at the surface. Massive *Alternaria* infections of ears (black ear rot) were reported in 1987, 1988, 1990, 1997, and 2020.

Saprotrophic populations of *Alternaria alternata* may develop pathogenicity and a toxin production ability as a result of mutations. Sometimes, pathotype *A. alternata* may switch between saprotrophic and parasitic forms, which is important to recognize in practice. The toxin production ability in *A. alternata* and the effect of the toxins themselves are reduced to the gradual suppression of resistance mechanisms in susceptible plants and the reduction of viability of stored seeds (Table 1).

The samples harvested in the dry year of 2010 showed a low *A*. *аlternata* population of 5 to 18% in grains after 8 years of storage, the germinating power being 20 to 72% and germinating rate, 35–95%. Thus, the seeds harvested in dry years preserved high viability even after 8 years of storage, as they were initially the least populated with the pathogen. Stored crops harvested in wet years are initially the most infected by *Alternaria*, and the viability of seeds is reduced after five years of storage, which is confirmed by the data collected in 2011 after 8 years of storage (Table 2).

It is worth noting that, according to our observations, the seed material with reduced viability should not be used for seed production since it accumulates a large number of genetic mutations. In seed production, mutations from saprotrophic types to specific plant pathogens producing host-specific toxins occur naturally and regularly regardless of the availability of susceptible varieties. These field observations explain sudden outbreaks of diseases caused by *A. alternata* in the seedlings of newly developed and introduced susceptible cultivars. 

In the case of spontaneous or intragenic mutagenesis, the energy sources are the bio-chemical changes in the cell. Spontaneous mutability is a consequence of metabolic disorders caused by the work of hydrolytic enzymes, amylases and proteases in conditions of excessive humidity and temperature [10].

Some researchers have also noted that sharp fluctuations in temperature, high humidity, and an increased background of natural radioactivity create specific conditions that contribute to the appearance of mutations [1,13]. We have [10] proved that during the action of a complex of certain temperatures and humidity on wheat plants leading to the manifestation of enzyme–mycotic depletion of seeds, the structure of proteins is destroyed, which is clearly revealed on proteinograms. Apparently, this is due to the release of proteases from vacuoles that destroy the structure of proteins.

We noted that the violation of the protein structure and enzyme affect the seeds’ sowing quality (Table 1 and Table 2). It should be noted that wheat plants resistant to the enzyme–mycotic depletion of seeds are characterized by anabolic processes, i.e., stable protein synthesis, while those susceptible to the enzyme–mycotic depletion of plant seeds show a shift in metabolic processes in favor of catabolic ones, characterized by the increased hydrolysis of proteins. It is interesting to note that the suppression of protein synthesis (and violations of their structure) in grain, which is accompanied by the enzyme–mycotic depletion of seeds in susceptible wheat varieties, causes the intensive development of phytopathogen on ears—pathogens of alternariosis, fusariosis and septoriosis. The data of the analysis of protein and enzymograms of proteins and amylase indicate that in the grain of plants exposed to increased moisture during filling and maturation, neither to the phase of full ripeness, nor 6 months after harvesting, the biochemical processes are not normalized and significant violations of the structures of proteins and enzymes are not restored [14]. This factor should be taken into account when developing long-term storage systems, since this indicates the internal causes of the loss of seed viability as a result of the effect of moisture and the manifestation of the enzyme stage of the enzyme–mycotic depletion of seeds during the filling and maturation of wheat.

The pathogen loses its specific virulence and toxin production ability in laboratory conditions. In addition, spontaneous toxin producing mutants could be observed in large populations of saprotrophic *A. alternata* under favorable conditions. Thus, specific virulence of *A. alternata* pathotypes seemingly depends on the ability to produce host-specific toxins, whose effects are reduced to the gradual suppression of resistance mechanisms in plants. The reduced viability of seeds with manifestations of parasitic fungal properties after five years of storage was recognized earlier [15,16]. 

A slight change in humidity and temperature conditions during long-term storage of seeds launches respiratory activity, hydrolytic enzyme activity, i.e., amylases and proteases, as well as the activity of *Alternaria* and other pathogens at the surface and inside caryopses. Sometimes, myco- and microbiota populations increase exponentially. 

Throughout the research period, the following cultivars showed tolerance to alternariosis of ears and grains: Zarya (k-49916), Yantarnaya 50 (k-54610), Nemchinovskaya 52 (k-59269), Nemchinovskaya 25, Мoсkoвсkая 39 (k-64160) from domestic Nemchinovskaya line, Mironovskaya 808 (k-43920), Rovenskaya 31, Odesskaya 51 (k-46620) from Ukraine; Bersee (k-40092, France); Compal (k-57585), Tukan (k-57579), Ibis (k-45335) from Germany; Kadav (k-57614), FM 187, PP 114-74 (k-57618) from Poland; Bocquiau (k-49824, Belgium).

### 2.6. Fusarium Ear Blight, Pathogens Such as Fusarium avenaceum (Fr.) Sacc., F. culmorum (W.G. Sm.) Sacc. etc.

Epiphytotics of *Fusarium* ear blight in the Moscow Region was reported twice in 50 years, specifically in winter triticale and spring wheat in 1979, and in winter triticale and spring and winter wheat in 1991. The disease was caused by biological injuries in standing crops, which is typical for the enzyme stage of EMDS. At the blooming and grain filling stages in July, 21 days with precipitation were reported, the monthly amount being 208.5 mm or 260% compared to the normal amount. August was warm and rainy (15 days), with precipitation of 180.8% of the normal amount, and relative humidity was 71–85%. The honeydew symptom was observed in male and female reproductive organs, i.e., anthers, rich in hydrolytic enzymes, carbohydrates, proteins, and lipids at the blooming phase. Humid weather with abundant precipitation intensified hydrolytic enzyme activity leading to biopolymer decomposition and honeydew development. The latter acted as a perfect substrate for phytopathogenic invasion. Fungi *Fusarium avenaceum* and *F. culmorum* were discovered by phytopathological examination. A total of 100% of wheat plants were covered by honeydew as early as the blooming phase, indicating the enzyme stage of EMDS. A week later, white, orange, and pink sporulation of the *Fusarium* species could be observed on the ears previously covered by honeydew (see Figure 4). 

Thus, the warm and wet conditions in August favored the replacement of *Alternaria* with *Fusarium* in this ecological niche. However, the study of a large wheat collection and resistance evaluation showed that each regional assortment included tolerant and highly tolerant samples (25–40% of infection) best suited for breeding. Those included: Zarya (k-49916), Nemchinovskaya 24 (k-65757), Nemchinovskaya 52 (k-59269), from the Moscow Region; Mironovskaya 808 (k-43920), Rovenskaya 31, Odesskaya 51 (k-46620) from Ukraine; Umanka (k-63041, Krasnodar Krai); Bersee (k-40092, France); Compal (k-57585), Ibis (k-45335) from Germany. It is worth mentioning that *Fusarium* infects winter rye, spring and winter triticale, and spring wheat in the Non-Black Earth Belt, while winter wheat is very rarely affected.

### 2.7. Septoria Leaf Blight, Pathogen Septoria tritici Rob. et Desm. and Ear Blight, Pathogen Stagonospora (syn. Septoria) Nodorum Berk

Throughout the research period, epiphytotics of septoria leaf and ear blight in the soft winter wheat from the world collection was only reported once in 1987. A massive outbreak of the enzyme stage of EMDS, i.e., biological injury of standing crops due to high air humidity and temperature was observed. Out of 141 samples studied, 15 demonstrated the highest resistance, with four of them showing an infection rate of 5% at the end of the vegetation period. These varieties, namely NS 2795 from the Netherlands, Rotor (k-57239), SR 8016 (k-57242) Amandus (k-57601) from Germany, had few septoria spots on middle leaves (Figure 5), while the upper leaves were left unscathed. Such cultivars and breeding lines as Bocquiau (k-49824), Rembrandt (k-49825) from Belgium; PP 114-74 (k-57618), ST 47/43, FM-187 from Poland; TAW 10929/65, Hadmerslebener 32798/78, Hadmerslebener 34496/75 (k-59706), Tukan (k-57579) from Germany showed the infection rate of about 10%. In two German samples, NS 2714 and Compal (k-57585), the rate was 15%. Reference cultivars Mironovskaya 808 and Zarya showed infection rates of 40–50%, but crop yields were still high at 650–740 g/m^2^. The other resistant samples had crop yields of 470–530 g/m^2^, except for German cultivars Tukan and Compal with 580–630 g/m^2^. As a result, these samples are of interest in terms of breeding for resistance to septoria leaf blight.

Winter wheat resistance to septoria ear blight was evaluated in the context of the EMDS enzyme stage at the end of the earing phase, upon achieving full milky-wax ripeness, and before harvesting (see Figure 6). Here, the samples with ear infection rates of 10–20% are of interest, including Fenman (k-57608, the United Kingdom), Liwilla (k-57580), PP 114-74 (k-57618), Gama (k-57581), Biala Kozsicka (k-56262), Rmo (k-55220), Kadav (k-57614) from Poland; Tukan (k-57579), Compal (k-57585), Rector (k-58304), Hadmerslebener 34496/75 (k-59706), Tabor (k-56907) from Germany; Bocquiau (k-49824), Rembrandt (k-49825) from Belgium; Trifolium 33 (k-56290, Denmark); Folke (k-58036), Sv Vg 73394 (k-56160), WW 26023 (k-58038), Salut (k-58035) from Sweden; Arina (k-57528, Switzerland); Zarya (k-49916), Yantarnaya 50 (k-54610), Nemchinovskaya 24 (k-65757) from the Moscow Region; Mironovskaya 808 (k-43920, Ukraine). Crop yields and 1000-grain weights of resistant samples matched those of the reference cultivar, i.e., 347–400 g/m^2^ and 37–48 g, respectively.

### 2.8. Germination in Standing Crops, Third EMDS Stage

Throughout the 50 year research of the winter wheat gene pool, germination in standing crops was reported three times in 1984, 1999, and 2003.

**In 1984**, a period of mass maturation from July 24 to July 30 was characterized by daily precipitation and high air temperatures of 23–25 °C. Resistance to germination in standing crops was evaluated in 472 samples and was reported in over 100 samples. The number of germinating grains was determined per 10 g, dry basis and converted into a percentage from the total number. Germination in standing crops was not observed in the following samples: Merkur (k-57010), Feldman (k-54135), Monopol (k-54706), 477/58 (k-53516), Ibis (k-45335), TAW 4229/74 (k-55944), Progress (k-49830) from Germany; ZG-2394/73 (k-54720), K-35A (k-51978) from Yugoslavia; Maris Hunstler (k-55230), Maris Settler (k-55229), Maris Durrin (k-55232), Standart Red (k-46000) from the United Kingdom; Donskaya polukarlikovaya (k-54647, Rostov Region). The samples that stood out in crop yields (400–500 g/m^2^) and 1000-grain weights (43.7–48.9 g) matched the results for the reference cultivar (405 g/m^2^ and 47.9 g). 

**The year 1999** was mostly dry, but a 13 day period of rainfalls in August during the full ripeness phase favored germination in standing crops in up to 2% of white kernel wheat samples out of 500 winter wheat samples.

**In 2003**, in August, at the beginning of the full ripeness stage was an 18 day period of frequent rainfalls with precipitation of 111.3 mm compared to the normal amount of 74 mm; the air temperature was +17.8 °С compared to the norm of +16.4 °С. Germination in standing crops was reported in over 200 samples from Western Europe and in some domestic samples, which hindered combined harvesting, and resowing was required. Resistance to germination in standing crops was demonstrated by the following samples: Raduga (k-50948, Moscow Region); Lgovskaya 77 (k-49917, Kursk Region); Soyuz 50 (k-49242, Belarus); Voronezhskaya 42 (k-49881), P-50-75 (k-50962), Chernozemka 56 (k-51735) from Voronezh Region; Lyutestsens 479 (k-15188, Omsk Region); WW 24089 (k-51803, Sweden); Jo 03022 (k-48345), Jo 3088 (k-51779) from Finland, C 564/69 (k-50684, Poland); Maris Nimrod (k-49846), Rothwell Senator (k-50762), Maris Ranger (k-50721), Compair (k-51913) from the United Kingdom; Starke II (k-51409, Sweden). Crop yields of these samples matched those of the reference cultivar and varied from 452 to 465 g/m^2^.

### 2.9. Immunological Study of Barley Yellow Dwarfluteo Virus (Hordeum Virus Nanescens Rademacheret Schwarz) in Winter Wheat

Barley yellow dwarf virus (BYDV) infects about 100 plant species from the grass family, including wheat, rye, barley, oat, corn, rice, and grasses. The disease, also called ‘yellow plague’, is the most harmful and common virus infecting cereals all over the world and causes epiphytotics, leading to major crop losses. Massive infections of winter wheat, spring barley, and spring oat were reported in 1990. Crop losses reached 90%, and some farms received crop yields of only 3–4 cwt/ha. Enzyme-linked immunoassay and indicator plant testing performed by A.I. Zezyukin, Ph.D., showed the presence of BYDV in infected plants. It was discovered that symptoms of the disease and the severity of their manifestations varied depending on cultivar sample, plant age, virus strain, and environmental conditions. It was also found that the leaf color in infected winter wheat plants changed from light-green to yellowish and from bright-yellow to purple. The infection of the winter wheat seedlings led to intensified tillering. The plants rarely reached the earing phase, and only a few ears on a few culms were observed. The crops were greatly reduced, and ears became sterile. The plants were not equal in height, with dwarf plants presented.

The following soft winter wheat varieties showed resistance and tolerance to the disease: Elmo (k-58051), Caldwell (k-58069), Adena (k-63966), Compton (k-59342, Keiser (k-60725) from the United States; Bezostaya 1 (k-42790, Krasnodar Krai); Yubileinaya 50 (k-59789, Kyrgyzstan); Albatros (k-58519), Odesskaya 51 (k-46620), Mironovskaya 808 (k-43920), Mironovskaya 61 (k-57671) from Ukraine. Winter wheat samples of variety *erythrospermum* Korn. were less affected by the virus.

## 3. Materials and Methods

Field trials of winter wheat were performed at the Federal Scientific Selection and Technology Center for Horticulture and Nursery, and phytopathological research, at the All-Russian Research Institute of Phytopathology and Moscow Timiryazev Agricultural Academy. 

The safety-duplicate collection of 2626 samples of soft winter wheat gathered by VIR and stored at the Department for the Gene Pool and Bioresources of Plants was used as the research material.

The climate in the Moscow Region is moderately continental and humid. The average annual precipitation is 450–800 mm. Accumulated temperatures above 10 °С decrease from 2100 °С in the east and south-east to 1900 °С in the north-west, and vegetation periods (above 10 °С) decrease accordingly from 140–145 to 120–125 days.

Availability of water and heat resources in the Moscow Region makes it possible to cultivate almost all temperate zone crops. The Stupino District is a part of the second (II) agroclimatic zone occupying the central part of the region and is included in subzone 11 а of soddy podzolic loam soils [17]. Soils freeze to the depths of 50–75 cm in open areas and 30–50 cm in sheltered areas. Full thawing of soil is usually expected from April 21 to 29. Soil tilth is achieved on May 20 in loam soils and on May 18 in sandy loam soils. The frost-free period usually lasts 120–135 days, which allows cultivated plants to achieve full ripeness. Permanent snow cover with an average depth of 35 cm forms between November 25 and December 2, and may persist up to 137–143 days. The hydrothermal index is 1.3–1.4. 

Winter wheat from the world collection was introduced into the scientific crop rotation (with a sowing density of 500 grains/m^2^) in the optimal seeding window from August 25 to 27 with black fallow crops. A SSFK-7M seeder was used in the plot areas of 2 m^2^. The NPK 68-60-30 mineral fertilizers were applied at the preplanting cultivation stage and N 50, as a secondary fertilizer in spring. The agricultural machinery was common for the studied region. The reference cultivars Mironovskaya 808, and in some years, Polukarlik 3, Zarya, Nemchinovskaya 52, and Moskovskaya 39 were planted with intervals of 10 and 50 samples respectively.

Every 5–8 years, the collection samples of winter wheat were replanted, and the seeds were stored in a special room at a temperature of 15–18 °C and an air humidity of 5–10%.

The wheat collection was studied according to the VIR Guidelines [18,19]; a unified classifier of Triticum L. was used [20]. Resistance to the enzyme–mycotic depletion of seeds was evaluated using original methods [8,10]. The classification of varieties by resistance groups was carried out according to the data of the average lesion of the variety: immune 0% lesion, resistance score 9; highly resistant, up to 10% lesion, resistance score 8; resistant 11–20% lesion, resistance score 7; medium-resistant 21–50% lesion, resistance score 5; susceptible lesion of 51–75%, resistance score 3; highly susceptible lesion of over 75%, resistance score 2; lesion of up to 100%, resistance score 1 [21]. The study of the genetic resources of grain crops for resistance to harmful organisms was carried out according to the following methods [22,23]: fusarium species on grain were determined according to V.I Bilai [24]; field assessment of resistance to root rot was carried out according to the methodological guidelines [25]; the yellow dwarfism virus of barley, by the enzyme immunoassay and indicator plants [26]; And diagnostics of plant resistance by the number of proteins–enzyme–electrophoresis, by the modified method of B.B.-O. Gromova [27].

## 4. Conclusions

The evaluation of resistance to biotic stress factors in the gene pool of winter wheat from the VIR world collection made it possible to identify valuable starting material for breeding. 

In this direction, the selection method of increasing resistance to biotic and abiotic conditions of agrocenoses is strategically more effective. This is possible through the creation of improved and new plant varieties that have a complex resistance to environmental stress factors [28]. Without solving the problem of complex resistance of plants to biotic and abiotic stresses, it is impossible to create forms that are resistant to any semi-parasitic or saprophytic pathogen (whether it is alternariasis, septoriosis or fusariosis). We isolated valuable samples from the gene pool of winter wheat, which are sources of resistance and tolerance to a complex variety of diseases.

It should be noted that the attempts to use the cultivars highly resistant to specific pathogen races in seed production were unsuccessful. The exclusion of the least virulent races from the farming ecosystem favors the emergence, selection and distribution of more aggressive races reduced crop yields and even losses of linear cultivars, as happened in the winter wheat varieties *Avrora* and *Kavkaz,* highly resistant to brown rust, in the third year of seed production [29]. 

We also confirmed that significant temperature fluctuations in the direction of increasing and high humidity contribute to the manifestation of a complex disease—the enzyme–mycotic depletion of seeds, which results in the destruction of the proteins structure (which are not restored) and the appearance of mutations.

Throughout the 50 year research period, the absence of disease in cereals was only reported once in 2014. 

The plasticity varieties Bezostaya 1 and Mironovskaya 808 (and spring variety Moskovskaya 35), susceptible to diseases, produced consistently high crop yields for over 40 years. A plasticity variety produces consistent annual crop yields in the cultivation area through its tolerance, rather than through its biological resistance, to stress factors [30]. 

A plant breeder working on an adaptive cultivar should primarily focus on tolerance rather than on high resistance. They should keep in mind that the viruses causing the asymptomatic course of the disease require additional instrumental methods to evaluate cultivar resistance.

In this respect, a strategy of combating diseases by tolerating pathogen development in cultivated plants without significant reduction in crop yields and quality seems to be a worthy option.

## Figures and Tables

**Figure 1 pathogens-10-00514-f001:**
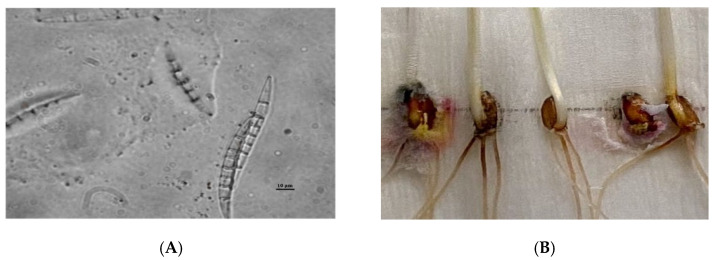
(**A**)—conidia of fungus *Microdochium nivale* (Fr.) Samuels and I.C.Hallett; (**B**)—symptoms of grain infection by *M. nivale*.

**Figure 2 pathogens-10-00514-f002:**
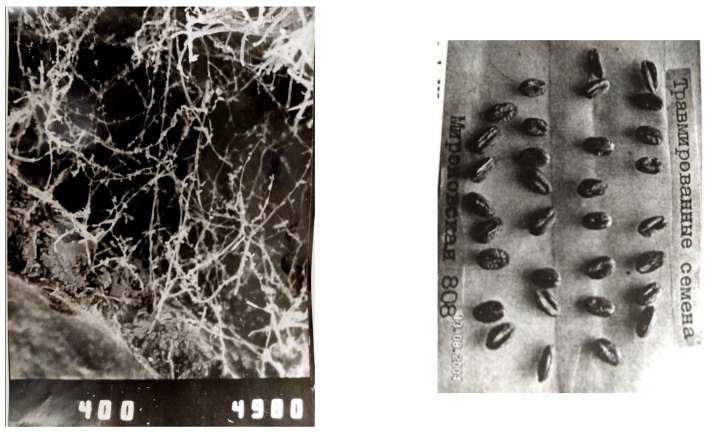
Biological injury of grains in the field at the stage of full ripeness as a result of the enzyme stage of EMDS: the introduction of a phytopathogenic infection through a hidden trauma on the left (×400), an open trauma on the right.

**Figure 3 pathogens-10-00514-f003:**
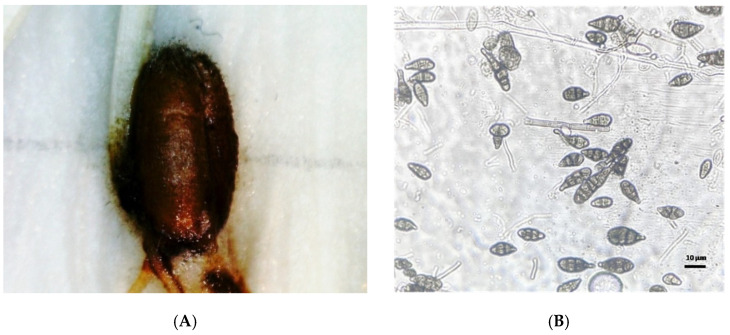
(**A**)—symptoms of *Alternaria alternata* (Fr.) Keissl. infection in grains; (**B**)—conidia of fungus *Alternaria alternata* (Fr.) Keissl.

**Figure 4 pathogens-10-00514-f004:**
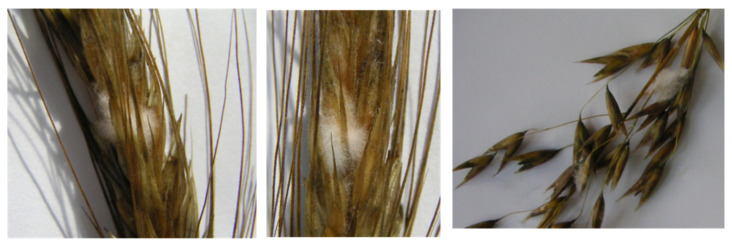
Fungus *Fusarium* spp. on honeydew in barley, triticale, and oat at blooming and milky ripeness phases.

**Figure 5 pathogens-10-00514-f005:**
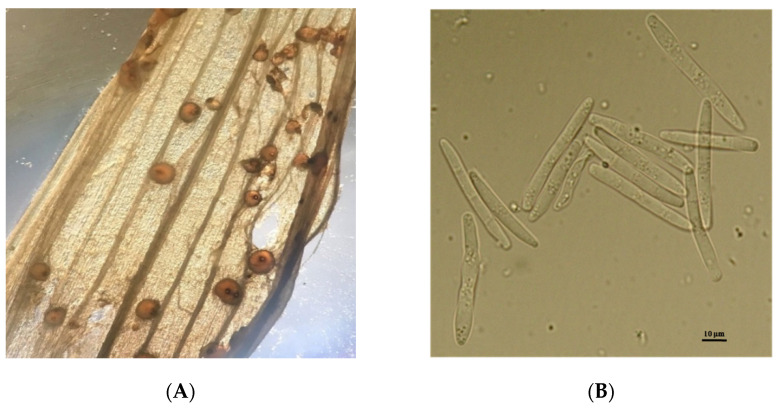
(**A**)—pycnidia of *Septoria tritici* Rob. et Desm on wheat leaves; (**B**)—conidia of fungus *Stagonospora nodorum* Berk.

**Figure 6 pathogens-10-00514-f006:**
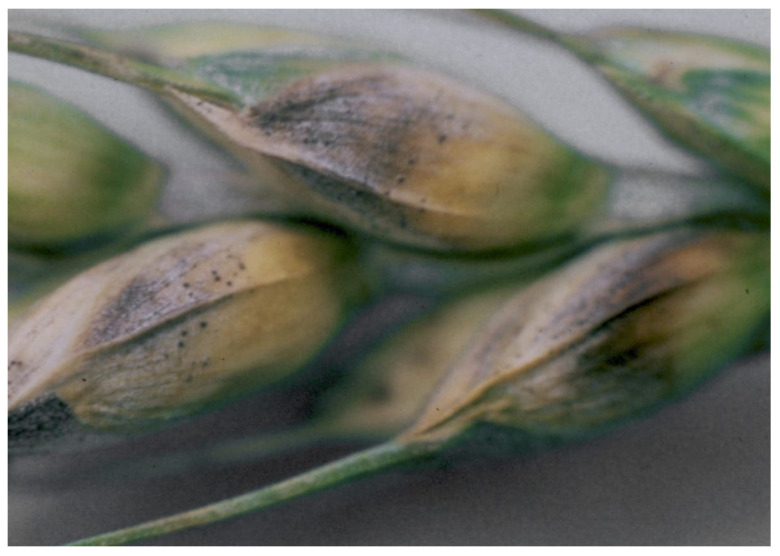
Symptoms of ear infection by *Stagonospora nodorum* Berk.

**Table 1 pathogens-10-00514-t001:** The effect of the pathogen *A. alternata* on the seeds’ viability of the 2010 harvest when they are sown after long-term storage after 8 years.

VIR Catalogue No.	Variety	Origin	Germinating Energy	Germinating Power	Infected Seeds,% (*A. Alternata*)	Crop Yield, g/m^2^
54668	Lutescens 444/73	Russia, Moscow Region.	24.00	35.00	5.00	570
54706	Monopol	Germany	59.00	80.00	11.00	510
55529	Tundra	the Netherlands	48.00	69.00	12.00	515
55801	Lutescens 12	Russia, the Kursk Region	44.00	78.00	9.00	600
55930	TAW 3295/73	Germany pre-1949	70.00	95.00	0.00	4225
55932	TAW 12181/72	Germany pre-1949	44.00	65.00	0.00	390
55933	TAW 13802/72	Germany pre-1949	31.00	52.00	0.00	510
55944	TAW 4229/74	Germany pre-1949	20.00	42.00	0.00	525
55971	L-1749	Russia, the Kursk Region	36.00	36.00	0.00	620
56339	Derwent	England	32.00	49.00	0.00	530
57221	Oberst	Germany	20.00	41.00	0.00	475
57222	Severin	Germany	45.00	60.00	0.00	620
57614	Kadav	Poland	42.00	65.00	0.00	540
58036	Folke	Sweden	72.00	84.00	0.00	505
58138	Ragnar	Sweden	40.00	64.00	15.00	420
58302	Falke	Germany	60.00	69.00	0.00	395
58305	Faras	Germany pre-1949	32.00	45.00	0.00	450
58367	SMH 1624	Poland	29.00	41.00	0.00	460
58831	Lutescens 398	Russia, Voronezh region	21.00	37.00	18.00	585
62274	Hankkijas Ilves	Finland	35.00	48.00	0.00	350
57009	Disponent	Germany	37.00	64.00	0.00	370

**Table 2 pathogens-10-00514-t002:** The effect of the pathogen *A. alternata* on the seeds’ viability of the 2011 harvest when they are sown after long-term storage after 8 years.

VIR Catalogue No.	Variety	Origin	Crop Yield, g/m^2^	1000-Grain Weight, g	Germinating Power	Germinating Energy	Infected Seeds %	
							*A. Alternata*	*Fusarium* sp.
5144	Kubb	Sweden	395	44	75.00	80.00	10.00	
9087	Banatka	Russia, Voronezh Region	535	24	8.00	8.00	97.00	
9681	-	Russia, Kaluga Region	467.5	29	1.00	4.00	92.00	
9695	-	Russia, Vladimir Region	322.5	40	39.00	45.00	88.00	
9696	-	Russia, Vladimir Region	465	27	48.00	52.00	40.00	
9762	-	Russia, Tver Region	360	33	12.00	12.00	24.00	
9765	Mestnaya	Russia, Pskov Region	512.5	30	4.00	12.00	99.00	
10010	Kraffts Verbesserter Siegerlaender	Germany pre-1949	382.5	33	0.00	0.00	95.00	
5183	Smoa	Sweden	347.5	37	20.00	32.00	37.00	
6258	Beal	Germany pre-1949	365	44	52.00	64.00	55.00	5.00
10267	-	Russia, Smolensk Region	297.5	38	8.00	12.00	85.00	9.00
10473	Tomskaya	Russia, the Tomsk Region	305	23	28.00	32.00	19.00	
10566	-	Russia, the Pskov Region	477.5	43	20.00	20.00	32.00	
10571	-	Russia, Leningrad Region	270	35	12.00	16.00	36.00	
6301	Roggen	Germany pre-1949	282.5	39	36.00	40.00	34.00	
6302	Roberts	Germany pre-1949	335	32	23.00	36.00	37.00	
13221	-	Russia, Tver Region	457.2	29	16.00	28.00	25.00	
13121	Tystofte Smaahvede II	Denmark	365	34	28.00	44.00	60.00	
15199	Moskovskaya 2470	Russia, Moscow Region	295	39	30.00	36.00	32.00	
19106	Kostromka	Russia, Tver Region	442.5	40	55.00	60.00	37.00	
22428	-	Russia, Tver Region	557.5	39	48.00	52.00	10.00	5.00
15197	Moskovskaya 2453	Russia, Moscow Region	390	37	3.00	8.00	57.00	
15198	Moskovskaya 2460	Russia, Moscow Region	317.5	35	4.00	6.00	49.00	3.00
19099	-	Russia, Tver Region	545	41	43.00	61.00	31.00	
19399	-	Russia, Krasnoyarsk Krai	385	32	0.00	4.00	99.00	
22391	Moskovskaya 2323	Russia, Moscow Region	357.5	42	6.00	17.00	36.00	
22418	Sandomirka	Russia, Orel Region	447.5	38	42.00	65.00	11.00	
25143	Ankar	Sweden	375	41	50.00	67.00	36.00	
25147	Kroontarwe	the Netherlands	382.5	47	0.00	0.00	95.00	
25153	W x EP	the Netherlands	590	44	35.00	40.00	22.00	
26140	Siegerlander	Germany pre-1949	357.5	42	25.00	38.00	15.00	
24500	Mikhailovka	Russia, Leningrad Region	402.5	37	2.00	5.00	42.00	
24923	-	Russia, Primorsky Krai	305	38	10.00	16.00	10.00	
50610	Hadmerslebener 25603/70	GDR pre-1990	532.5	40	81.00	84.00	4.00	
50672	Hadmerslebener 42498/71	GDR pre-1990	675	46	56.00	68.00	33.00	
19108	-	Russia, Tver Region	375	39	20.00	28.00	35.00	
51423	977	Russia, Tambov Region	585	49	8.00	24.00	61.00	
59546	General	Germany	535	50	20.00	44.00	29.00	
60304	TAW 8913/74	Germany pre-1990	770	50	24.00	40.00	52.00	
26228	Bensings Trotzkopf	Germany pre-1949	407.5	46	80.00	87.00	8.00	

## Data Availability

For the first time, an analysis of a fifty-year study of the gene pool of winter wheat for resistance to various diseases is presented.

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
