# Peer review of "Gene Pool of Winter Wheat from the World Collection of N.I. Vavilov Institute of Plant Industry (VIR) for Biotic Stress Resistance"

_pathogens, 2021, doi:10.3390/pathogens10050514_

Round 1
Reviewer 1 Report
This manuscript provides an extensive analysis, using historical data, of the winter wheat germplasm contained within the VIR world collection. The authors use phytopathological phenotyping data to help identify genotypes which may offer valuable germplasm for future breeding programs. Because the goal is to identify breeding materials, this work should cite other studies that use molecular tools to explore the VIR germplasm. For instance, Riaz et al. 2018 used select lines from the VIR collection to create a diversity panel for genome-wide association studies (GWAS) to identify leaf rust resistance loci. This type of study takes advantage of the diversity of materials contained within the VIR collection and provides an efficient way to identify novel disease resistance alleles.
Riaz, A., Athiyannan, N., Periyannan, S. K., Afanasenko, O., Mitrofanova, O. P., Platz, G. J., ... & Voss-Fels, K. P. (2018). Unlocking new alleles for leaf rust resistance in the Vavilov wheat collection. Theoretical and applied genetics, 131(1), 127-144.
The introduction and conclusion sections should be expanded to further discuss the value of such a germplasm collection for plant breeders and how the current study can be used by others to identify materials for future molecular analysis, marker assisted breeding, etc.
Author Response
The first reviewer wanted us to complete the introduction and conclusion. We did and highlighted it in blue.
Reviewer 2 Report
The manuscript “Gene pool of winter wheat from the world collection of N.I. Vavilov Institute of Plant Industry (VIR) for biotic stress resistance breeding” is an interesting summary of 50 years of observations of numerous accessions coming from all over the world. The authors focused on fungal diseases of cereals and a single viral one caused by barley yellow dwarf virus. Admittedly the presented studies are not innovative but they are valuable in another way. Many years of observations allow to select those varieties that show resistance to diseases and which can be used in breeding as sources of resistance.
The title of the manuscript should be expanded because the authors presented results concerning not only wheat but also (to a small extent) barley, triticale, and oat. Maybe “winter wheat” should be replaced with “cereals”.
The methodology must be more detailed. Admittedly, the authors quote a few literature items but most of them is written in Russian and is not available for an average reader. The weather conditions at the place of cultivation and the method of cultivation have been described in detail but there is not any information how the resistance/tolerance/susceptibility of tested accessions were determined. What scale of diseases symptoms was used for determining a degree of infestation with fungal diseases? How the immunoassay for detecting BYDV was performed? What conditions the seeds were stored in? What was temperature? humidity? How often individual accessions were sowed and regenerated?
The authors use unclear kind of transcript, for example: 30…40 g (line 115). I suppose that means: a range from 30 to 40 g but according to me it should be written with a hyphen: 30 - 40 g. There is a lot such mistakes in the text.
Scale bars should be included on the microscopic photos (Figures 1A, 3A, 5B)
Figure 1 B: The subtitle of this photo does not explain what is presented on it. Figure 2 on the left showing hidden injury of grains must be removed because the same photo has been presented yet in the published article: S. K. Temirbekova, A. V. Ovsyankina, N. E. Ionova, T. D. Cheremisova, Y. V. Afanasyeva, O. P. Mitrofanova, M. H. Al-Azawi Nagham, Enzymatic activity in the resistance stress of winter wheat from different sources in the non-black land of the center of Russian Federation, Plant Archives Vol. 19 No. 1, 2019 pp. 1653-1658 as figure 1 on the page 1657.
Tables 1 and 2 have the same title.
Lines 216 - 218. The author wrote: “It is worth noting that, according to our observations, the seed material with reduced viability should not be used for seed production, since it accumulates a large number of genetic mutations.” In my opinion, the authors’ conclusion that the seeds accumulates genetic mutations is unfounded because they did not make any molecular analyses which could reveal mutations.
And a few small remarks: Line 22: There is “powdery milde wattacks” while it should be ”powdery mildew attacks”
Line 53: There is “to isolate”. Maybe “to select” would be better.
Lines 80 - 90: There is “g/m2” while "2" should be written in top index.
Author Response
The second reviewer wished:
1. expand the methodology. We have expanded the methodology, it is highlighted in blue.
2. Add the scale bars in Fig. 1a, 3a, and 5b. We added a scale ruler.
3. we made a subtitle in Fig. 1b. Completed.
3. Figure 2 was replaced with another picture. completed.
4. In Tables 1 and 2-the names were written more accurately. Completed.
5. line 291-323 gave a reasoned explanation for the genetic mutation. Completed.
6. line 32-defeat powdery mildew. Completed.
7. line 115-126 - edits made. Completed.